

# Bacterial diversity of herbal rhizospheric soils in Ordos desert steppes under different degradation gradients

Yuefeng Guo[1], Dan Zhang[1] and Wei Qi[2]

[1] College of Desert Control Science and Engineering, Inner Mongolia Agricultural University, Hohhot, Inner Mongolia, Asia, China
[2] Inner Mongolia Autonomous Region Water Conservancy Development Center, Hohot, Inner Mongolia, Asia, China

Corresponding author
Yuefeng Guo,
2022202250037@emails.imau.edu.cn

## ABSTRACT

**Objectives**. This study explored the effects of different degradation gradients on bacterial diversity in the rhizospheric soils of herb plants.

**Methods**. The alpha diversity, species composition and correlations of bacterial communities in the rhizospheric soils of herb plants were studied using metagenomics 16SrDNA gene high-throughput sequencing.

**Results**. The diversity of bacterial communities in the rhizospheric soils of herb plants differed during the degradation of desert steppes. An analysis of bacterial community alpha diversity indices showed the bacterial diversity and species evenness of rhizospheric soils were best in moderately degraded desert steppes. Among all samples, a total of 43 phyla, 133 classes, 261 orders, 421 families, 802 genera and 1,129 species were detected. At the phylum level, the predominant bacterial phyla were: Actinobacteria, Proteobacteria, Acidobacteria, Gemmatimonadetes, Chloroflexi, Planctomycetes and Bacteroidetes. At the genus level, the predominant bacterial genera were: RB41, Sphingomonas, WD2101_soil_group_unclassified, Pseudomonas and Actinomyces. The relative abundance of unknown genera was very large, which deserves further research. At the phylum and genus levels, the species abundance levels under slight and moderate degradation were significantly higher than those under extreme degradation. Correlation network diagrams showed there were many nodes in both slightly deteriorated and moderately deteriorated soils, and the node proportions were large and mostly positively correlated. These results indicate the bacterial communities in rhizospheric soils under slight or moderate deterioration are relatively stable. The rhizospheric soil microbes of desert steppes can form a stable network structure, allowing them to adequately respond to environmental conditions.

**Conclusions**. The bacterial communities in the rhizospheric soils of herb plants differ between different degradation gradients. The species number, abundance and diversity of bacterial communities in rhizospheric soils are not directly correlated with degree of degradation. The abundance, species diversity and species abundance of bacterial communities in the rhizospheric soils of moderately degraded desert steppes are the highest and most stable. The soil bacterial diversity is lowest in severely degraded desert steppes.

## INTRODUCTION

Desert steppes are the most xeric type of grasslands. Small xeric and fascicular herbs that are often mixed with numerous small xeric suffrutexes, and constitute stable predominant layers in communities are the predominant plants in stable desert steppe communities. Desert steppes are geographically located in inner continent areas and have an annual precipitation of ≤200 mm. Grassland degradation is a major form of desertification caused by both natural and human factors, and is a retrogressive succession of grassland ecosystems (*Liu, Lu & Chen, 2011*). Desert steppes are ecologically unique because of their arid habitats and unique vegetation compositions and are fragile to human interference (*Wei et al., 2013*). The large desert steppes found in Inner Mongolia are important to stockbreeding production, human life and the ecological safety of both Inner Mongolia and China as a whole (*Ma, 2008*). Because of the importance of desert steppes, their ecological environmental status has attracted much attention (*Ma, 2008*). Population increases and the heavy demand for animal by-products have resulted in excessive grazing in grasslands (*Wang et al., 2021*) leading to grassland degradation throughout Northern China. As a result, the capacity, soil fertility, and proportion of high-quality pasture in grasslands have all decreased, hindering the development of grassland eco-construction and slowing stockbreeding in China.

Research is increasing on desert steppe deterioration. *Zhang et al. (2022)* investigated the relationship between the biomass of four dominant plant species and grazing intensity in the desert steppes of Inner Mongolia and found that as grazing intensity increased, the aboveground biomass and underground biomass of desert steppe communities both declined. *Wu et al. (2010)* used PCR-DGGE technology to analyze the soil bacterial community structure of desert grasslands with different degrees of degradation and found that the soil bacterial community structure differed significantly by degree of degradation. *Han (2019)* found the trait indices of plants decreased as grassland deterioration increased.

Soil microbes can recover degraded soils and promote ecosystem recovery (*Yu et al., 2021*). The structure and function of soil microbial communities are the key indices of grassland degradation (*Yu et al., 2021*). Rhizosphere soil is the most active area of biogeochemical cycling around roots, the site of interaction between soil, plant roots and microorganisms, and the gateway of various material circulation and energy flow, which plays an important role in the dynamic distribution and circulation of nutrients in ecosystems and the interaction between plant species (*Curl & Truelove, 1986*; *Jones, 1998*; *Weller, 1988*). A global meta-analysis showed that mild to moderate grazing did not affect the biomass of soil microbial, bacterial or fungal communities, but severe grazing significantly reduced the total biomass of soil microbial, bacterial and fungal communities (*Zhao et al., 2017*). Another global meta-analysis demonstrated that mild to moderate grazing improved bacterial community abundance, and severe grazing decreased bacterial community abundance, while fungal community abundance largely increased under mild grazing and dropped under moderate to severe grazing (*Wang & Tang, 2019*).

A Chinese study found that compared with undegraded grasslands, the quantities of bacteria and fungi rose significantly under mild grassland deterioration, but significantly

declined under moderate to severe deterioration, while the quantity of actinomycetes increased with the level of deterioration (*Cai et al., 2004*). However, another previous study found that as the grassland degradation degree intensified, quantities of soil bacteria, fungi and actinomycetes gradually decreased, reaching the smallest amount in severely-degraded grasslands (*Wen et al., 2014*). *Li et al. (2016)* found that compared with undegraded meadows, the abundance and Shannon diversity index of soil bacterial and fungal communities under severe deterioration significantly increased, but moderate deterioration did not significantly affect the diversity of bacterial communities. *Hu et al. (2014)* found that microbial species homogeneity, abundance and community diversity of moderately-deteriorated alpine meadows were the highest, showing that microbial community composition is more complex.

These existing findings on the changing rules of soil microbial diversity and community composition during grassland degradation are inconsistent. There is not currently a published study on bacterial diversity in the rhizospheric soils of herbal plants in desert steppes under different degradation gradients degree. This study sought to identify whether root soil bacterial communities differ by level of desert steppe degradation. Identifying how the community diversity and composition of soil bacteria changes during the grassland degradation process will guide the formulation of relevant recovery measures (*Luo et al., 2022*). This study used high-throughput sequencing to study the root soils of herb plants under different degradation gradients in the desert steppes of Ordos. We speculate that the rhizospheric soils of desert steppes under different degeneration levels also obey the intermediate disturbance hypothesis, meaning the species number, abundance and diversity of bacterial communities in the rhizospheric soils under moderate deterioration are better than those under slight, severe or extreme deterioration, and are the most stable. The findings from this study will help uncover how degradation gradients impact the diversity of bacterial communities in the rhizospheric soils of herb plants and provide a basis for the ecological restoration of desert steppes in Inner Mongolia.

## MATERIALS & METHODS

### The study area

This study was carried out in the monitoring sites of Naogaodaigacha grassland in Aerbasisumu, Etuoke, Ordos, Inner Mongolia. The coordinates of the center of the study field are E107°34′29.48′ and N39°35′9.27′. The altitude of the study area ranges from 1,100 to 1,500 m. This area has a temperate continental monsoon climate with four distinct seasons, rich solar radiation, dry and rainless weather, high levels of evaporation and sandy wind. The perennial mean temperature is 6.4 °C. The annual average sunshine duration is 3,000 h, with a 122-day frost-free period, 250 mm of precipitation, and 3,000 mm of evaporation. Most annual precipitation occurs between July and September. Etuoke is perennially dry with grassland desert vegetation. Grassland degradation has intensified and the proportion of desert grassland is rising annually. Local soils are thin and sandy. The main soil types are brown calcic soil and gray desert soil. The sample we have taken this time is brown calcic soil. The dominant species of vegetation communities include *Stipa breviflora*, *Artemisia frigida*, and *Ceratoides latens* (*Ren & Ge, 1992*).

## Sampling

The experiments and field investigation were conducted in July 2022 in the collective Naogaodaigacha grassland of Aerbasisumu, Etuoke banner, Ordos, Inner Mongolia. The different functions of different grassland areas have resulted in different levels of degradation, with severe deterioration observed around drinking sites. A natural grazing succession gradient has formed in each grassland, with ground features and plant community compositions differing significantly based on distance from the drinking site. This has created a radial space transformation pattern centered around the draining site, with degradation decreasing successively from the center to the circumference. Grassland degradation near the drinking water point is a retrograde succession of grassland under the action of external and internal factors. The degree of grassland degradation is positively correlated with the distance from the drinking water point, which is a process from quantitative change to qualitative change. Animal trampling plays a leading role in the degradation of grasslands near drinking water point. With the aggravation of the degradation degree, the environment deteriorates further, from normal grassland to the degree degradation stage is a quantitative change process. With the aggravation of degradation degree, it leads to severe degradation of grassland, and then to extreme degradation.

For this study, five radial belts centered around the drinking site were set in the experimental field, with each belt 6 km long. On each belt, 36 sampling sites (1 m × 1 m) were set by degradation degree, and 12 groups (three sites in each group) were formed. The most distant group from the drinking site was marked W1, and the nearest group was marked W12. The fenced control area was used as the control group (W13).

The vegetation succession sequence of degradation in the desert steppe was: *Stipa breviflora → Stipa breviflora + Artemisia frigida → Artemisia frigida + Cleistogenes songorica* (*Liu et al., 2002*). Based on the grassland degradation classification theory by *Li (1997)*, "Principles of Classification of Degraded Grassland" (*Zhao & Qi, 1987*) formulated by the Inner Mongolia Grassland Survey and Design Institute, the perennial experience of relevant experts as well as local meteorological data, degradation level in this study was mainly based on the aboveground biomass of plant communities, vegetation coverage and indicator plants. The specific levels of degradation were defined, as follows:

1. Light degradation. The plant community coverage was decreased by 20–30%, with the grass group mostly maintaining its original appearance, and total grassland production decreasing by less than 30%. The original dominant plant production accounted for 40%–60% of the total grassland production. The surface was heavily eroded by wind, and there was slight sand covering or gravel fossilization.

2. Moderate degradation. The plant community coverage was decreased by 31–45%, and the grass community was sparse and low, with the dominant species being *Convolvulus albion* and one to two-year-old miscellaneous grass. The total yield of the grassland was decreased by 30%–60%, with the yield of the original dominant plants accounting for 10%–40%, and the yield of various degraded indicator plants accounting for 15%–40% of total grassland production. The surface wind erosion was severe, and the surface was moderately covered with sand or gravel.

**Table 1** Description of sampling sites.

| Plot number | Coverage/% | Above-ground biomass/g· m$^{-2}$ | Dominant species | Degree of degeneration |
|---|---|---|---|---|
| W1 | 20.14 ± 3.56[ab] | 31.93 ± 3.56[b] | *Stipa breviflora, Eragrostis pilosa, Parthenocissus tricuspidata* | Light |
| W2 | 19.83 ± 6.53[ab] | 30.55 ± 7.64[b] | *S. breviflora, Plantago minuta* | Light |
| W3 | 18.94 ± 8.79[ab] | 28.86 ± 5.78[b] | *S. breviflora, E. pilosa, Allium mongolicum* | Light |
| W4 | 18.33 ± 4.93[a] | 28.73 ± 5.65[a] | *S. breviflora, E. pilosa* | Moderate |
| W5 | 18.02 ± 5.76[a] | 27.63 ± 4.87[a] | *S. breviflora, P. tricuspidata, Setaria viridis* | Moderate |
| W6 | 17.39 ± 3.78[a] | 24.98 ± 5.41[a] | *S. breviflora, Agropyron mongolicum* | Moderate |
| W7 | 17.09 ± 4.86[a] | 24.44 ± 1.36[ab] | *S. breviflora, S. viridis* | Severe |
| W8 | 16.34 ± 3.76[a] | 21.71 ± 5.56[ab] | *S. breviflora, Eragrostis minor* | Severe |
| W9 | 15.97 ± 4.39[a] | 20.39 ± 7.35[ab] | *S. breviflora, Artemisia frigida, E. minor* | Severe |
| W10 | 10.07 ± 3.47[c] | 13.36 ± 5.35[c] | *S. breviflora, A. frigida, Cleistogenes songorica,* | Extreme |
| W11 | 10.85 ± 3.76[c] | 13.05 ± 3.57[c] | *S. breviflora, A. frigida, Peganum harmala* | Extreme |
| W12 | 9.98 ± 3.89[c] | 12.39 ± 1.79[c] | *S. breviflora, A. frigida, P. harmala* | Extreme |
| W13 | 22.09 ± 3.87[b] | 43.67 ± 6.98[b] | *Ceratoides latens, S. breviflora, E. pilosa* | No |

**Notes.**
Different lowercase letters indicate significant differences between groups ($P < 0.05$).

3. Severe degradation. The plant community coverage was decreased by 46–60%, with the established and dominant plants in the grassland accounting for less than 10% of the total yield of the grassland. The invasive plant *Camelaria camelaria* became dominant in the grassland, and the yield of various degraded indicator plants accounted for more than 40% of the total yield of the grassland. Extreme wind erosion was present on the surface, and there was a large amount of gravel.

4. Extreme degradation. The plant community coverage decreased significantly (>60%). The vegetation and soil conditions further deteriorated with serious grassland erosion, and the overall productivity and ecological functioning of the area declined.

According to the above classification criteria, we divided the 13 groups in the plots into the four degradation degrees: lightly degraded (W1, W2, W3), moderately degraded (W4, W5, W6), severely degraded (W7, W8, W9), extremely degraded (W10, W11, W12), and a control (W13). Soils were sampled using a five-point sampling method. First, the aboveground plants, dry branches and fallen leaves on the ground surface were removed. The soils within 0–10 cm in each plot were sampled, and the rhizospheric soils of the herb plants were obtained using a shake-off method (*Riley & Barber, 1970*). The soils from the groups with the same mark were put onto the same oilcloth (disinfected on site with 75% alcohol before use) and mixed. A total of 39 samples, each weighing about 500 grams, were put into one aseptic bag, which was then put into a dry ice foam box for transport and then stored in a freezer at −80 °C before microbe experiments. The sampling sites are shown in Fig. 1. The basic information of the sample plots is listed in Table 1.

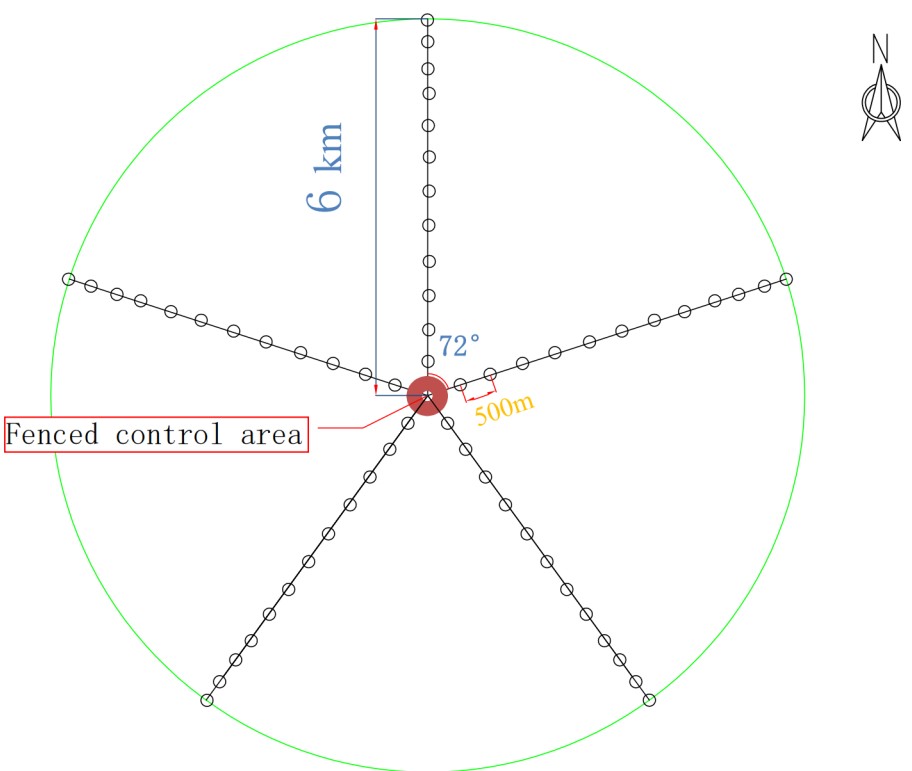

**Figure 1** Experimental design route map.

## Methods

### Extraction of microbial total DNA

Total DNA from the bacterial groups of different sources was extracted using cetyl trimethylammonium bromide (CTAB). DNA quality was then detected *via* agarose gel electrophoretic analysis, and DNA was quantified using ultraviolet spectrophotometry.

### PCR amplification and product quantification

Regions V3–V4 were amplified using 16SrDNA gene primers 341F (5′-CCT ACGGGNGGCWGCAG-3′) and 805R (5′-GACTACHVGGGTATCTAATCC-3′). The PCR products were corroborated *via* 2% agarose gel electrophoresis. Ultrapure water was used throughout the DNA extraction, and the possibility of false positive PCR results as negative control was excluded. The PCR products were purified using AMPureXTbeads (Beckman Coulter Genomics, Danvers, MA, USA) and quantified using Qubit (Invitrogen, Waltham, MA, USA).

### Recovery and purification of amplification products

The PCR amplification products were detected using 2% agarose gel electrophoresis and recovered using an AMPureXTbeads recovery kit.

### Quantification, mixing, and machinery detection of PCR amplification products

The purified PCR products were monitored using an Agilent2100 biological analyzer and an Illumina library quantitative kit. The qualified library concentration was above 2 nM. The qualified libraries were diluted on a gradient and then denaturalized with NaOH into single strands before sequencing on the machine (the index sequence was not repeated). Paired end sequencing of 2 × 250 bp was conducted *via* envoy NovaSeq6000 sequencing, and the corresponding reagent was 500 cycles.

## Biological information analysis flowchart and data processing
### Data splitting

The paired end data from the sequencing were split according to barcode to remove the sequences of joints and barcodes.

### Data splicing and filtering

The specific steps were taken:

1. The primer sequences and balance base sequences were removed from the raw data (software: Cutadapt (v1.9), parameter: '-gR1-GR2-n1-O17-m100').

2. Paired end reads were spliced and combined into a longer tag based on the overlap zone (software: FLASH (v1.2.8), parameter: '-m10-M100-x0.25-t1-z').

3. The quality of the sequencing reads was scanned using the window method (default of scanned window = 100 bp); when the average quality in the window was lower than 20, the part of the read from the start to the 3′end in the window was cut off (software: fqtrim, parameter: '-P33-w100-q20-l100-m5-p1-V-otrim.fastq.gz').

4. The cut sequences shorter than 100 bp were removed.

5. The cut sequences that contained more than 5% N (the uncertainty blur base) were removed.

6. Chimera sequences were removed (software: Vsearch (v2.3.4), default parameters).

### Denoising of DADA2

Length filtering and denoising were conducted by transferring DADA2 *via* qiimedada2denoise-paired. The sequences and abundance of ASV (feature) were obtained, and then single ASVs (ASV (feature) with total sequence number of 1; default operation) were removed (*Callahan et al., 2016*).

### Diversity analysis and species annotation

Alpha diversity was analyzed according to the ASV (feature) sequence and ASV (feature) abundance table using seven indices, including observed_species, Shannon, Simpson, chao1, goods_coverage, and Pielou_e. Species were annotated according to the ASV (feature) sequence files using SILVA (Release138, https://github.com/QIIME2/q2-feature-classifier) and the NT-16S database. The abundance of each species in each sample was summarized according to the ASV (feature) abundance table. The confidence interval of annotation was 0.7.

### Data processing and variance analysis of root soil sequencing

Differences between groups were compared based on species abundance data. Data were reorganized using Excel 2019, and the variance analysis was conducted on the DPS data processing system (*Gan, Cheng & Li, 2008*). Plotting was finished on Origin2018 and R-3.4.4.

### Alpha diversity analysis

Alpha diversity refers to the diversity in a specific environment or ecosystem and reflects species richness and evenness as well as sequencing depth. Alpha diversity is analyzed using the chao1, observed_species, goods_coverage, Shannon, Simpson and Pielou-e indices.

Chao1 and observed_species reflect the number of species in a community. Goods_coverage is the microbial coverage rate, with larger values indicating a lower probability of not detecting new species. Shannon's index is derived from information entropy, and a larger index means heavier uncertainty. A community with heavier uncertainty involves more unknown factors and is more diverse. Simpson's index varies within 0–1. When a community only has one species, the Simpson's index is the smallest (0), or has the least diversity. When the species are infinitely abundant (the highest abundance) and the numbers of all species are consistent (high evenness), Simpson's index is the largest (1). The Pielou-e index is also called Shannon's evenness index, with a larger value indicating higher evenness.

## RESULTS AND ANALYSIS

### Alpha diversity analysis of soil bacterial communities under different degradation gradients

The histograms of the relative abundance of microbial communities showed that each sample was composed of highly-expressed species. The composition and expressions of within-group species and the expressions of intergroup species were also observed. Results of 16S rDNA sequencing are shown below.

On Fig. 2A, each circle represents one group, and the overlap between two circles shows the number of shared ASVs between the two groups. The part of the circles that do not overlap show the ASVs exclusive to each group. Groups W1, W2, W3 and W13 contained 1,295; 912; 1,135 and 661 ASVs, respectively, with 45 shared ASVs. On Fig. 2B, groups W4, W5, W6 and W13 contained 863; 920; 1,228 and 615 ASVs, respectively, with 71 shared ASVs. On Fig. 2C, groups W7, W8, W9 and W13 contained 1,120; 991; 1,361 and 587 ASVs, respectively, with 57 shared ASVs. On Fig. 2D, groups W10, W11, W12 and W13 contained 1,548; 1,332; 303 and 562 ASVs, respectively, with 20 shared ASVs.

Analysis and detection showed the diversity of soil bacterial communities differed by degradation gradient. As can be seen from Table 2 and Figs. 3–4, observed_species and chao1 were not significantly different between groups W1–W9 and W13, but were significantly larger in these ten groups than in groups W10–W12. Observed_species were ranked, as follows: W9 > W7 > W6 > W5 > W4 > W8 > W > W3 > W2 > W13 > W12 > W10 > W11 (Fig. 3B), indicating the bacterial community diversity was higher in groups W1–W9 and W13, and lower in groups W10–W12. Shannon's index (Fig. 4) was
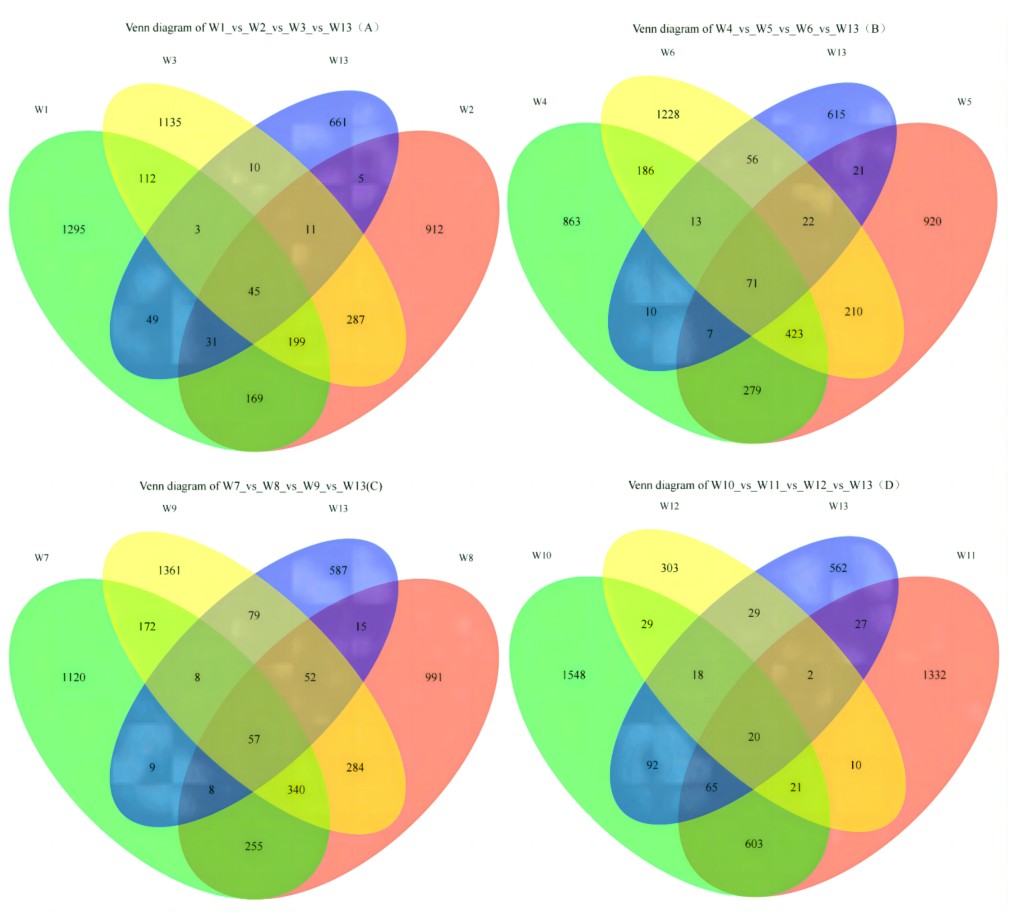

**Figure 2  Venn diagrams of ASVs.**

significantly higher in groups W4–W6 (10.44,10.46,10.3) than in the other groups, but was lowest in groups W10–W12 (9.78, 8.75, 7.54, respectively), which were significantly lower than in the other four groups. The Pielou-e index differed significantly between groups W10–W12, W13 and the other nine groups ($P < 0.05$). It was the highest in groups W4–W6, and the lowest in groups W10–W12, indicating the homogeneity of bacterial communities was the lowest in groups W10–W12 and the highest in groups W4–W6. Simpson's index did not significantly differ between groups W4–W6 and the other 10 groups, or between group W13 and groups W4–W12; it was 1.00 in groups W4–W12 and 0.99 in group W13.

## Species compositional analysis of soil bacterial communities under different degradation gradients

Among all samples, a total of 43 phyla, 133 classes, 261 orders, 421 families, 802 genera and 1,129 species were detected. Differences in the predominant bacterial phyla or genera were analyzed ($P < 0.05$) (Fig. 5). The predominant bacteria of groups W10–12 were

**Table 2  Alpha diversity index of each sample.**

|     | observed_species | Shannon | chao1 | pielou_e | Simpson |
|-----|------------------|---------|-------|----------|---------|
| W1  | $1{,}952.00 \pm 57.00^{ab}$ | $10.08 \pm 0.22^{b}$ | $1{,}956.69 \pm 70.79^{a}$ | $0.92 \pm 0.0058^{a}$ | $1 \pm 0^{a}$ |
| W2  | $1{,}849.00 \pm 58.60^{ab}$ | $9.83 \pm 0.14^{b}$ | $1{,}852.95 \pm 101.34^{a}$ | $0.92 \pm 0.0153^{a}$ | $1 \pm 0^{a}$ |
| W3  | $1{,}901.00 \pm 37.00^{ab}$ | $10.04 \pm 0.34^{b}$ | $1{,}903.92 \pm 104.23^{a}$ | $0.91 \pm 0.0058^{a}$ | $1 \pm 0^{a}$ |
| W4  | $2{,}000.00 \pm 97^{a}$ | $10.44 \pm 0.29^{a}$ | $2{,}084.51 \pm 62.33^{a}$ | $0.92 \pm 0.01^{a}$ | $1 \pm 0^{a}$ |
| W5  | $2{,}070.00 \pm 60.00^{a}$ | $10.46 \pm 0.59^{a}$ | $2{,}207.61 \pm 54.44^{a}$ | $0.92 \pm 0.0058^{a}$ | $1 \pm 0^{a}$ |
| W6  | $2{,}198.00 \pm 81.00^{a}$ | $10.30 \pm 0.23^{a}$ | $2{,}006.86 \pm 34.13^{a}$ | $0.92 \pm 0.0115^{a}$ | $1 \pm 0^{a}$ |
| W7  | $2{,}344.00 \pm 161.00^{a}$ | $10.12 \pm 0.04^{ab}$ | $2{,}407.88 \pm 213.06^{a}$ | $0.92 \pm 0^{a}$ | $1 \pm 0^{a}$ |
| W8  | $1{,}965.00 \pm 227^{a}$ | $10.13 \pm 0.01^{ab}$ | $1{,}973.91 \pm 100.75^{a}$ | $0.92 \pm 0.01^{a}$ | $1 \pm 0^{a}$ |
| W9  | $2{,}381.00 \pm 223^{a}$ | $10.09 \pm 0.045^{ab}$ | $2{,}358.36 \pm 275.18^{a}$ | $0.92 \pm 0.0058^{a}$ | $1 \pm 0^{a}$ |
| W10 | $1{,}659.00 \pm 167^{c}$ | $9.78 \pm 0.76^{c}$ | $815.00 \pm 245.26^{b}$ | $0.84 \pm 0.0611^{ab}$ | $0.99 \pm 0.0058^{ab}$ |
| W11 | $432.00 \pm 189^{c}$ | $8.75 \pm 0.32^{c}$ | $1{,}659.00 \pm 157.68^{b}$ | $0.81 \pm 0.0058^{ab}$ | $0.99 \pm 0.0034^{ab}$ |
| W12 | $813.00 \pm 134^{c}$ | $7.54 \pm 0.43^{c}$ | $432.60 \pm 276.34^{b}$ | $0.80 \pm 0.085^{ab}$ | $0.99 \pm 0.0087^{ab}$ |
| W13 | $1{,}795 \pm 229.00^{b}$ | $10.06 \pm 1.03^{b}$ | $1{,}804.74 \pm 135.01^{ab}$ | $0.87 \pm 0.0252^{b}$ | $0.99 \pm 0^{b}$ |

**Notes.**
Different lowercase letters indicate significant differences between groups ($P < 0.05$).

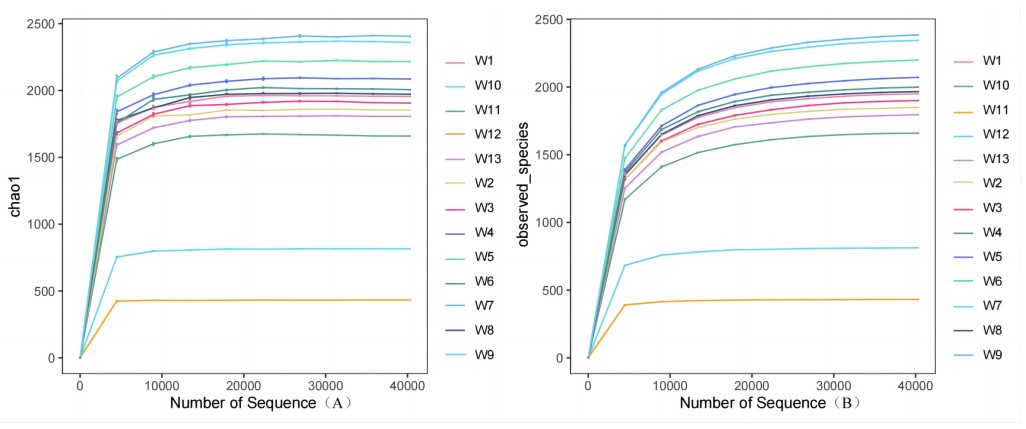

**Figure 3  Plots of Alpha diversity chao1 (A), observed_species (B).**

significantly different from the other 10 groups, but no significant differences were found among these 10 groups.

At the phylum level, the top ten predominant bacterial phyla ranked by relative abundance were: Actinobacteriota, Proteobacteria, Acidobacteriota, Gemmatimonadota, Chloroflexi, Planctomycetota, Bacteroidota, Firmicutes, Myxococcota and Verrucomicro-biota, which together accounted for more than 92.45% of the total bacterial phyla. After averaging the data from three replication groups, the relative abundance of Actinobacteria was the largest in groups W4, W5 and W6 (39.08%, 38.94%, 36.18%, respectively) and the lowest in group W12 (16.03%). The relative abundance of Proteobacteria was the largest in groups W4, W5 and W6 (35.67%, 27.39%, 29.86%, respectively) and the lowest in group W12 (22.65%). The relative abundance of Acidobacteria was the largest in group
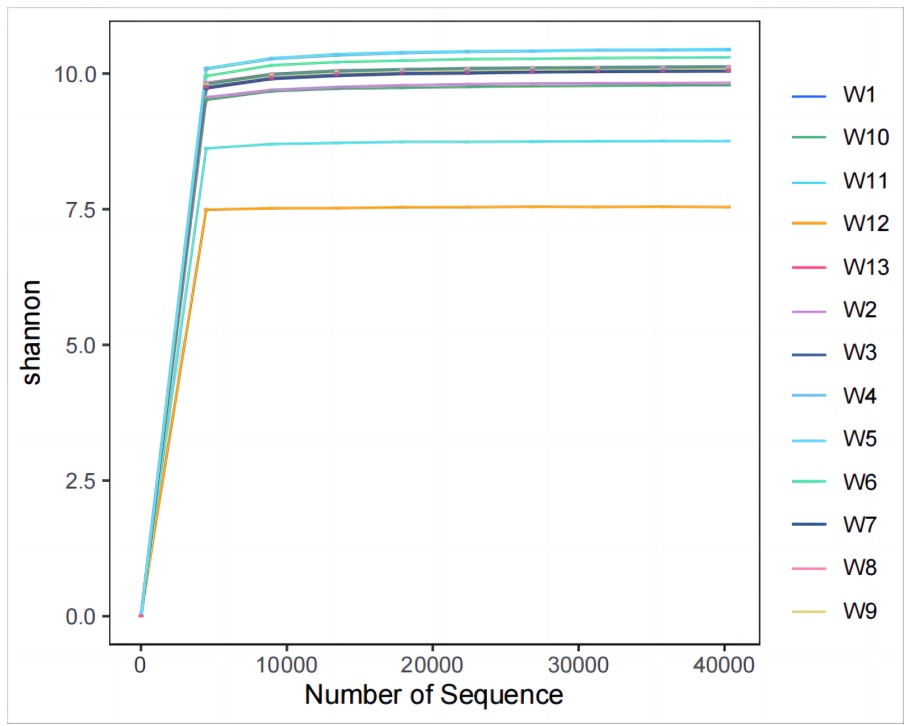

**Figure 4** Plots of Alpha diversity evenness index.

W4 (14.25%) and the lowest in groups W10, W11 and W12 (9.6%, 8.7% and 9.24%, respectively).

At the genus level, the top eight predominant bacterial genera ranked by relative abundance were: RB41, Actinobacteriota_unclassified, Sphingomonas, WD2101_soil_group_unclassified, Vicinamibacteraceae_unclassified, Gemmatimonadaceae_unclassified, Pseudonocardia and Gaiellales_unclassified, which together accounted for more than 13.06% of the total bacterial genera. The proportion of unknown genera was the largest (>42.39%). After averaging the data from three replication groups, RB41 accounted for the largest proportion of known genera, and was the most abundant in groups W4, W5 and W6 (6.75%, 6.52% and 6.35%, respectively). The species richness in groups W4–W6 was higher than in groups W10–W12.

The top 30 communities were then clustered at the phylum level and genus level according to the abundance distribution of taxons or based on the between-sample similarity. The taxons and samples were both ranked according to the clustering results and displayed on heatmaps. Clustering differentiates the high- and low-abundance taxons and reflects the similarity or differences among samples at the phylum or genus level by color gradient. At the phylum level, the abundance of Acidobacteriota, Gemmatimonadota, Abditibacteriota, Fibrobacterota and Deferribacterota was the highest in groups W4–W6 (12.67, 7.38, 0.22, 0.027 and 0.022, respectively, Fig. 6) and the lowest in groups W10–W12 (11.21, 4.98, 0.12, 0.017 and 0.003, respectively). At the genus level, the average abundance of RB41, Sphingomonas, Solirubrobacter and Actinophytocola was the highest in groups

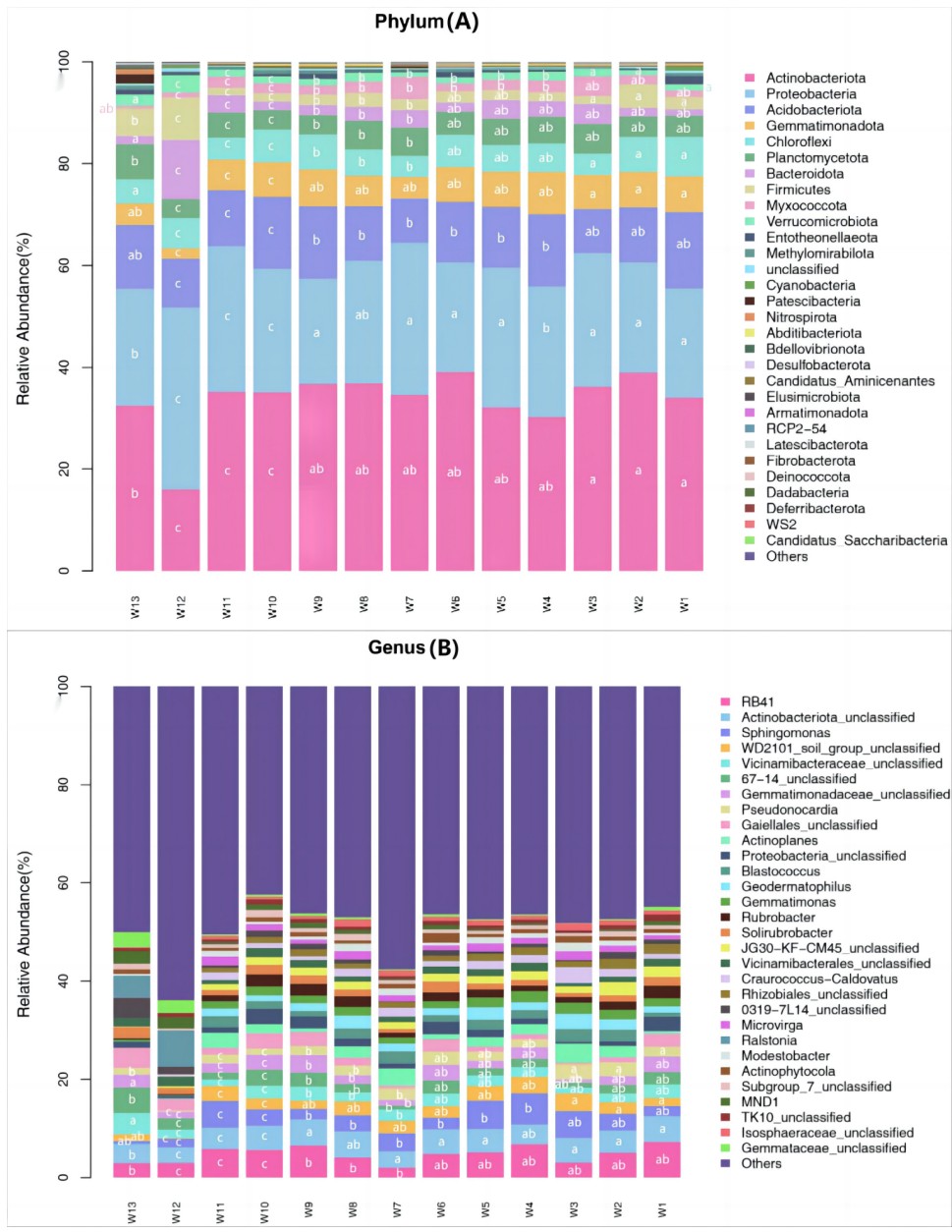

**Figure 5** Species abundance of phylum taxonomy of soil bacterial communities under different degradation gradients.

W4–W6 (5.58, 4.82, 1.54 and 1.16, respectively) and the lowest in groups W10–W12 (4.21, 3.02, 1.30 and 0.52, respectively).

An ASV feature phylogenetic tree was built of the alignment results of multiple ASV feature sequences. The top 50 species at the genus level, ranked by abundance, were plotted in Fig. 7, with branches representing different classes at the genus level. Different genera in the same color indicate they belong to the same phylum. A closer distance between two species means their phyletic evolution is closer. As shown in Fig. 7, light pink, light blue,
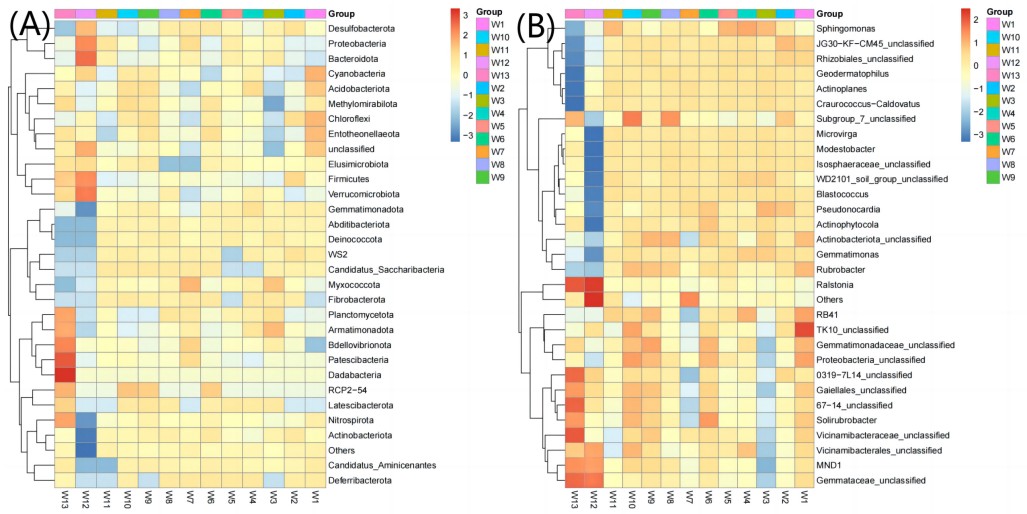

**Figure 6** Heatmap analysis of phylum taxonomy of soil bacterial communities under different degradation gradients.

and dark pink stand for Proteobacteria (three genera), Actinobacteria (18 genera) and Acidobacteria (six genera), respectively. The evolutionary distance of each phylum was very large.

## Correlation analysis of soil bacterial communities under different degradation gradients

Figure 8A shows the top five phyla and their abundance levels, and Fig. 8B shows the groups. A larger width means a higher abundance, and a smaller width indicates a lower abundance. The relative abundance (TOP5) from the interaction between soil bacteria at the phylum level differed by degradation gradient (Fig. 8A). The relative abundance of *Actinobacteria* was the largest in all 13 groups, with a mean abundance of 36.38, 33.82, 36.05 and 28.77 in groups W1–W3, W4–W6 and W7–W9, respectively. The mean abundance of *Actinobacteria* in group W4-W6 was the highest, which was higher than the control group, W13 (32.48). The largest proportion of *Proteobacteria* was found in groups W5, W7, W11 and W12 (27.39, 29.86, 28.56 and 35.67, respectively). The largest proportion of *Acidobacteriota* was found in group W1 (15.00) and the smallest proportion of *Gemmatimonadota* was found in group W12 (2.08). At the genus level (Fig. 8B), RB41 and *Sphingomonas* had the largest proportions in groups W4–W6 (5.58 and 4.82, respectively). The relative abundance of bacterial phyla (TOP5) was small in group W12 under different degradation gradients. The relative abundance (TOP5) of soil bacteria at the genus level differed by degradation gradient, with the lowest levels seen in group W12.

Associations among species as well as species composition were both studied using microecology experiments. In the Spearman analysis, the microbial abundance of the top 30 species were calculated at the genus level and correlations and significance level between paired predominant communities were determined and correlation network

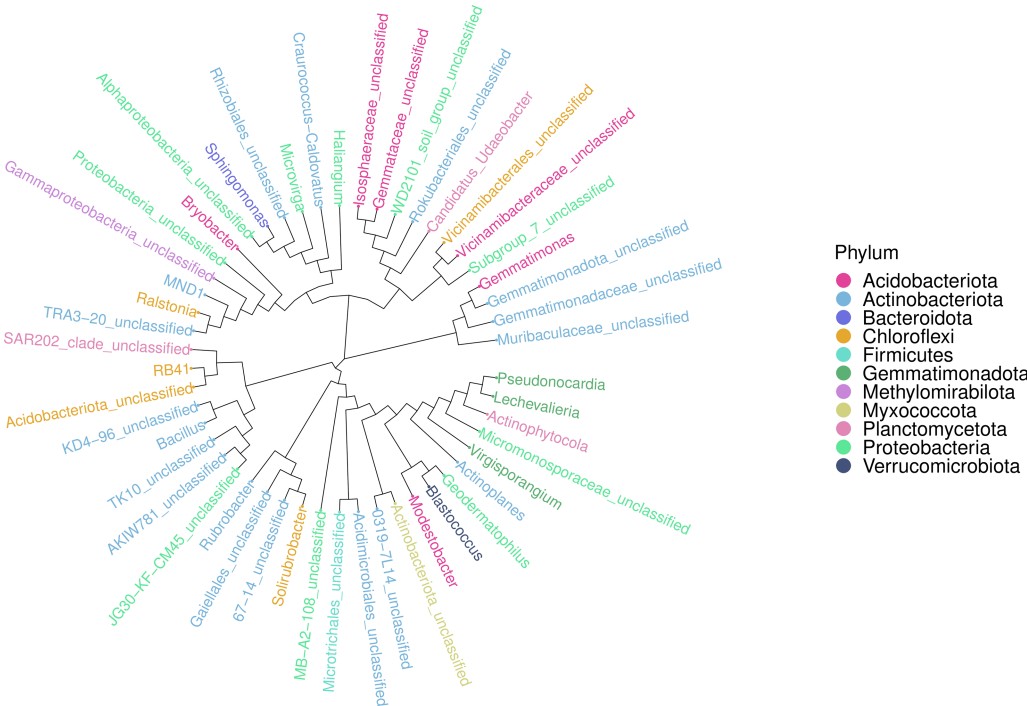

**Figure 7** Sequence phylogenetic tree of soil bacterial communities ASV features at different degradation gradients.

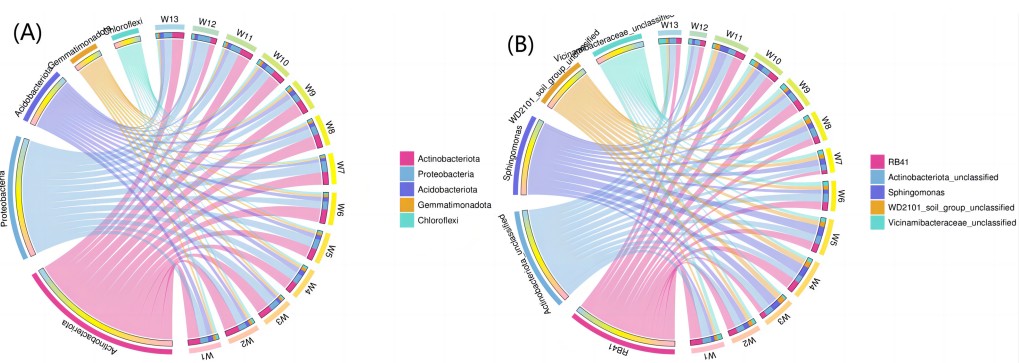

**Figure 8** Circles of phylum of soil bacterial communities under different degradation gradients.

diagrams were plotted. The network diagram (Fig. 9) had a total of 18 nodes, indicating the predominant genera, with four colors representing the phylum of the corresponding species. The connection between nodes indicates a correlation between two genera, using the correlation coefficient |rho| > 0.8. A thicker line indicates a stronger correlation. The solid and dashed lines indicate positive and negative correlations, respectively. *Acidobacteriota* showed a negative correlation, had nine nodes, and was correlated with the most genus

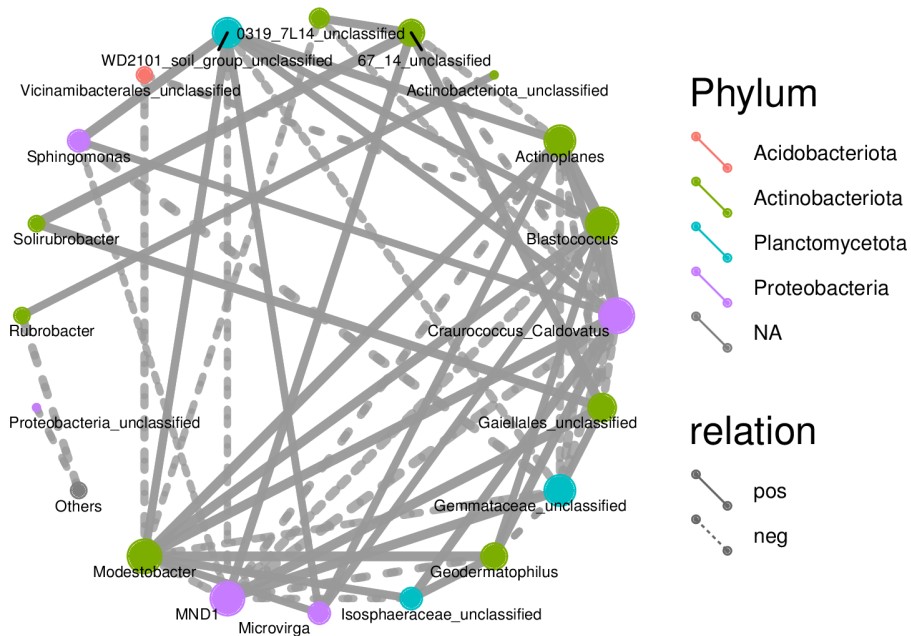

**Figure 9** Correlation networks of soil bacterial communities at different degradation gradients.

nodes. *Proteobacteria* had the smallest node and was associated with the least number of other genus nodes (only four nodes).

## DISCUSSION

The alpha diversity analysis showed the diversity of soil bacterial communities differed among different deterioration gradients. The observed_species, chao1, Shannon index, and Pielou index of the extreme degradation groups were the lowest, followed by the control group, in which the alpha index, abundance and evenness were the lowest. The four indices of the moderate degradation group were larger than the light degradation group, the severe degradation group, and the control group indicating the alpha diversity, abundance and evenness were the highest in the moderate degradation group. There was no significant difference noted between the light degradation group and the moderate degradation group. Retrograde succession of grassland ecosystems inhibits microbial growth and propagation, and further leads to a decline in microbial quantity (*Zhang et al., 2018*). Similar to the findings of a previous meta-analysis, this study showed that severe grazing significantly decreased total biomass and abundance of soil microbial, bacterial and fungal communities (*Zhao et al., 2017*).

During grassland degradation, the predominant microbial phyla remain unchanged, but the relative abundance changes. The rhizospheric soil bacterial community species composition analysis detected a total of 43 phyla, 133 classes, 261 orders, 421 families, 802 genera and 1,129 species in all samples. At the phylum level, the top ten predominant bacterial phyla, ranked by relative abundance, were: Actinobacteria, Proteobacteria, Acidobacteria, Gemmatimonadetes, Chloroflexi, Planctomycetes,

Bacteroidetes, Firmicutes, Myxococcota and Verrucomicrobia. Actinomycetales, the first dominant phylum, can decompose various organic matter (*Chen, 2003*) and plays a contributive role in both natural matter circulation and in the biological processing of sewage and organic solid waste. Actinomycetales can also promote soils to form aggregate structures and improve soil quality (*Chen, 2003*). Proteobacteria, the second dominant phylum, is the most universal soil bacterial phylum in the world and is closely related to carbon source utilization (*Spain, Krumholz & Elshahed, 2009*). Proteobacteria mainly consists of gram-negative bacteria that are responsible for nitrogen fixation, thereby increasing soil nitrogen concentrations (*Song, Wang & Wei, 2020*; *Chaudhry et al., 2012*; *Huang et al., 2012*). A higher proportion of Proteobacteria in soils increases plant growth and soil fertility preservation (*Chaudhry et al., 2012*). Proteobacteria, Actinobacteria, Acidobacteria, Bacteroidetes, Chloroflexi, and Planctomycetes are also the dominant bacterial phyla in the soil bacterial communities of deteriorated grasslands in the Hulun Lake Protection Zone of Inner Mongolia. The relative abundance of Proteobacteria and Bacteroidetes gradually decrease as grassland degradation intensifies, but the relative abundance of Actinobacteria, Chloroflexi and Planctomycetes increase (*Yu et al., 2021*). The predominant microbial species of grasslands remain basically consistent, but the relative abundance levels differ between different grasslands. Grassland degradation not only changes the species composition of microbial communities, but also affects the diversity and composition of microbial communities. These results indicate that the degradation of grasslands significantly affects the species composition of rhizospheric soil bacterial communities.

Unknown genera accounted for a large proportion of total genera, indicating more research is needed at the genus level. Among the known genera, the predominant genera were RB41, Sphingomonas, WD2101_soil_group_unclassified, Pseudomonas and Actinomyces, with RB41 having the largest proportion. The proportion of Actinomyces in the control group was 0. Pseudomonas belonging to P-accumulating organisms include Acinetobacter and Bacillus (*Shen et al., 2023*). Pseudomonas can degrade antibiotics (*Feng, 2022*). The species abundance of rhizospheric soil bacteria in slightly or moderately degraded soils is higher than that of extremely degraded soil.

Network diagrams of correlation analyses provide new clues for studying the complex relations among rhizospheric soil microbial communities (*Deng et al., 2012*). Previous research indicates that interactions among microbes indicate the response of microbial communities to external disturbances (*Wei et al., 2019*). The correlation network diagrams in this study show the rhizospheric soils under slight or moderate degradation contain many nodes that are complexly connected, indicating the rhizospheric soil microbes under slight or moderate degradation form a complex network and reproduce complex interactions and community structures. This study can find more changes of the resource ecological niche of soil microbes in desert steppes. A network with lower connectivity can be more easily interfered than a complex network with higher connectivity (*Santolini & Barabási, 2018*). A larger node means more associations and the proportion of important connecting points is large, indicating the rhizospheric soil bacterial communities under slight or moderate

degradation are relatively stable. Moreover, the connecting nodes critically contribute to the stability and connecting lines of the whole network (*Guimerà & Nunes Amaral, 2005*).

The species number, abundance and diversity of bacterial communities in rhizospheric soils are not directly correlated with level of deterioration. The abundance, species diversity and species abundance of bacterial communities in the rhizospheric soils of moderately deteriorated steppes are the highest and most stable. It is speculated that bacterial diversity is related to grassland degradation and that appropriate interference can improve the species diversity of rhizospheric soil bacterial communities in grasslands (*Dieison et al., 2020*), which is line with the intermediate disturbance hypothesis (*DiTommaso & Aarssen, 1989*). The results of this study provide a basis for the future ecological restoration of desert steppes in Inner Mongolia.

## CONCLUSIONS

The diversity of bacterial communities in the rhizospheric soils of herb plants differed during the degradation process of desert steppes. An analysis of bacterial community alpha diversity indices showed the bacterial alliance diversity and evenness of rhizospheric soils were the lowest in both undegraded desert steppes and extremely degraded desert steppes. The species numbers were similar between the undegraded group and the extremely degraded group, and the abundance and diversity under moderately degraded soils were higher than those of undegraded or extremely degraded soils.

Among all samples, a total of 43 phyla, 133 classes, 261 orders, 421 families, 802 genera and 1,129 species were detected. At the phylum level, the main predominant bacterial phyla were: Actinobacteria, Proteobacteria, Acidobacteria, Gemmatimonadetes, Chloroflexi, Planctomycetes and Bacteroidetes. At the genus level, the predominant bacterial genera were RB41, Sphingomonas, WD2101_soil_group_unclassified, Pseudomonas and Actinomyces. The relative abundance of unknown genera was very large, which deserves further research. At the phylum and genus levels, the species abundance levels under slight and moderate degradation were significantly higher than those under extreme degradation.

Correlation network diagrams showed the numbers of nodes were large in both slightly degraded soils and moderately degraded soils, and the node proportions were large and mostly positively correlated. These results indicate the bacterial communities in rhizospheric soils under slight or moderate degradation are relatively stable. The rhizospheric soil microbes of steppes can form a stable network structure, which can enhance the ability of rhizospheric soil microbes to adequately respond to this degradation.

### Funding

This work was supported by the Ordos Science and Technology Cooperation Key Project (2021EEDSCXQDFZ011); the Inner Mongolian Autonomous Region Directly Affiliated Universities Basic Scientific Research Operating Expenses Project (BR22-15-01); the Autonomous Region Application Technology Research and Development Fund Program

(2021GG0085); and the Natural Science Foundation of Inner Mongolian Autonomous Region (2022MS03029). The funders had no role in study design, data collection and analysis, decision to publish, or preparation of the manuscript.

### Grant Disclosures

The following grant information was disclosed by the authors:
The Ordos Science and Technology Cooperation Key Project: 2021EEDSCXQDFZ011.
The Inner Mongolian Autonomous Region Directly Affiliated Universities Basic Scientific Research Operating Expenses Project: BR22-15-01.
The Autonomous Region Application Technology Research and Development Fund Program: 2021GG0085.
The Natural Science Foundation of Inner Mongolian Autonomous Region: 2022MS03029.

### Competing Interests

The authors declare there are no competing interests.

### Author Contributions

- Yuefeng Guo conceived and designed the experiments, performed the experiments, prepared figures and/or tables, authored or reviewed drafts of the article, and approved the final draft.
- Dan Zhang conceived and designed the experiments, performed the experiments, analyzed the data, prepared figures and/or tables, and approved the final draft.
- Wei Qi conceived and designed the experiments, prepared figures and/or tables, authored or reviewed drafts of the article, and approved the final draft.

### DNA Deposition

The following information was supplied regarding the deposition of DNA sequences:
The raw sequence data reported in this paper have been deposited in the Genome Sequence Archive (Genomics, Proteomics & Bioinformatics, 2021) in the National Genomics Data Center (Nucleic Acids Res, 2022), China National Center for Bioinformation/Beijing Institute of Genomics, Chinese Academy of Sciences (GSA: CRA012135) that are publicly accessible at https://ngdc.cncb.ac.cn/gsa/browse/CRA012135.

### Data Availability

All the data used in the paper is in the Supplemental File.

### Supplemental Information

Supplemental information for this article can be found online at http://dx.doi.org/10.7717/peerj.16289#supplemental-information.

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
