# Peer review of "Bacterial diversity of herbal rhizospheric soils in Ordos desert steppes under different degradation gradients"

_PeerJ, doi:10.7717/peerj.16289_

## Round 0.1 · original submission · Major Revisions

Dear Authors,

Thank you for submitting your work to PeerJ.

The two reviewers have now completed their inspections and submitted their reports for your study. However, one of the reviewers recommended rejection while the other reviewer also found multiple errors in the manuscript. By going through their reports, I would like to provide an opportunity for you to revise your manuscript thoroughly based on their suggestions and comments through point-by-point responses. However, please be aware that this is not a guarantee for the acceptance of your study. Your revised manuscript will be undergoing a second-round of review by experts after the re-submission.

Kind regards,
Prof. Liang Wang, PhD

Reviewer 1 ·

Basic reporting

The manuscript entitled “Bacterial diversity of herbal rhizospheric soils in desert steppes under different degradation gradients” is an interesting manuscript which focusing on rhizosphere bacterial community in desert steppes, Inner Mongolia. While the topic is interesting, the beginning of introduction does not quite sets scene nicely about history of desert research. The introduction did not cover background research on microbial community which is the main focus of this study (see major concerns below). Materials and Methods are not adequate. Results and Discussion will need to be revised when the major concerns have been addressed. Overall, I think that the manuscript idea is interesting and will be valuable for the scientific community. However, additional work is needed to address major concerns and improve language and several parts of this manuscript to be more transparency and justified.

Experimental design

Line 99 – 111: Please add references to all the details provided here.

Line 120 - 127: How does the degradation level was measured? Currently, the details provided are not sufficient for other research to follow/repeat the experiment (Major concern). How can other researcher identify light vs. moderate degraded? How many samples were collected in total?

Line 123: 13rd – 13th

Line 139: Total DNA from the bacterial groups – please add more details how bacterial DNA was extracted without other microorganisms. (Major concern)

Line 144: Bacterial community composition was measured in LC Sciences – samples were sequenced by LC Sciences?

Line 159-160: double-end – paired end

Line 161-162: Please add reference.

Line 165: Please add references and version of databases

Line 168-169: Please provide data publicly and add details about the sequences such as sequence depth- what are the min, mean, max number of reads?

Need to provide citations for all software used in statistical analyses (Major concern)

Validity of the findings

Need statistical analysis to support statements in this section (Major concern)

Need to check figures number (Major concern)

Line 177-178: can be removed.

Line 180: Refer to figure 2?

Line 189-203: should not be in results section

Line 205: The – the

Line 207 – 214: Need to provide statistical analysis results (Major concern)

Line 236: TOP – top

Line 250 - 256: should not be in results section

Additional comments

Title
Indication of sampling site in the title would be crucial.

Abstract
Line 16: Alpha - alpha

Line 21: Alpha - alpha

Line 23: degradation levels are different from table 1.

Introduction

The introduction need to focus much more on bacterial community. This is not the first study to investigate desert microbial community. What have been found in previous research? Currently, there’s not much details about bacterial community (Major concern). Please take a look at previous research that have been conducted elsewhere (Major concern). Additionally, many references were added without discussing how they related to this study while additional references will needed to be added in many statements. Lastly, hypotheses were not provided (Major concern).

Line 52-54: Please add references

Line 57: Need to remove extra space

Line 63-70: It is not meaningful to list other studies without trying to connect to this study and provide details of what they found and how their results related to this study.

Line 71-72: Please add references

Line 84-85: Reference, why does this matter?

Line 93: “the changing rules” – not sure what this means.

Discussion
Line 294 - 300: not a discussion

Line 306: Please provide reference

Line 344 – 347: contradiction – how could this be new ideas when there’s a theory since 1989?

Overall, this study needs to focus more on the microbes. What are the key bacteria in each degradation level and also bacterial networks. What do they do? Why would they be essential? Try to be specific and compare to other studies.

Reviewer 2 ·

Basic reporting

The English is somewhat difficult to follow at times, with some examples of incorrect words used (expressed for OTUs, etc.)
The literature cited appears to be mainly geared towards studies in Asia. Given the location of the study, this makes sense, but given that steppes are widespread, it seems likely that studies from Africa and the Americas could also be relevant.

Experimental design

There are no hypotheses or questions clearly stated in the introduction or abstract. The methods section is quite vague, particularly regarding defining degradation, sequencing, bioinformatics and the analyses performed. Statistics are missing altogether.

Validity of the findings

The raw sequences do not appear to have been deposited into a publicly available database. Omics studio requires a login to access. Sequences should be deposited in GenBank, MGRAST or similar.
Given the lack of statistics, the conclusions cannot be assessed.

Additional comments

Understanding the effects of degradation on the soil microbiome is critical in light of increasing anthropogenic pressures on ecosystems, particularly fragile systems such as steppes. The current study unfortunately does not include a sufficient description of 1) how degradation was quantified for low-extreme, 2) how sequences were prepared for high throughput sequencing, 3) bioinformatics, including quality filtering, merging, etc. 4) statistics to support conclusions. No statistics were reported in the manuscript.

Specific comments:

The introduction would benefit from being split into subsections

Need to include a definition of steppe (e.g., dry, cold grassland) as well as degradation in the introduction.

Multiple statements begin with author names (lines 62-71), which is an unusual approach. Perhaps reduce this to the key studies that really underlie the hypotheses being tested in this study. In addition, these citations are only listed as studying a particular component of microbes in steppe ecosystems without giving the results of said studies. What did Wu et al. (2010) find in their analysis using DGGE?

Line 92. 16S sequencing is generally described as 16S rRNA gene sequencing rather than rDNA.

Line 93. What is meant by changing rules?

Need to better describe how the degrees of degradation were measured. There is not a clear explanation of sampling, or why distance from the watering hole would be linked to degree of degradation. Is there other data from prior studies that would support that this is a good analogue for degradation? What is a crawling area?

Line 134. At -80, this would be considered a freezer not refrigerator.

For Fig 1, given that this is an English language journal, all test should be English. Not sure what the middle text says. Watering hole?

Need more explanation of DNA extraction – was the CTAB method combined with phenol chloroform? How were nucleic acids purified?

Need to provide thermocycling conditions, annealing temperature, number of cycles for PCR1 and PCR2, polymerase used.

Why was PE 250 bp sequencing used for this region? The amplicons would be 461bp, which means that the overlap is quite small (~10nt). Why was the NovaSeq used rather than MiSeq?

Trimmomatic is a tool for quality filtering of fastq files – what is meant by purified? What were the quality scores used for filtering? Were reads trimmed? More information is needed for the bioinformatics portion. What percentage of forward and reverse reads were merged? What was the allowed merge overlap used with flash? What was used to classify sequences to taxonomy – what is a Bayesian classifier like RDP, or kmer like SINTAX, which version of SILVA was used as the reference database, what was the cutoff for successful classification? How were non-bacterial sequences (e.g., mitochondria and chloroplast) identified and removed?

Missing references for QIIME2, DADA2, SILVA, R.

Where were the sequences deposited?

What is meant by highly expressed? Since this is DNA and 16S, expression is not being measured. Please clarify.

It is not necessary to define diversity in this context, but rather which measures were used. Why were so many used? Should have a hypothesis for using the different metrics. Can also just called Chao and observed_species richness for simplicity.

How was difference in sampling depth dealt with? Rarefaction, standardization?

Line 197-198. This is an inaccurate statement regarding uncertainty and diversity.

There are no statistics performed on the results – not PERMANOVA to compare species composition, no ANOVA for diversity measures, therefore statements like this site was the lowest cannot be accurately evaluated. Similarly, the network (correlation) analysis has no statistical tests, and so the figures are interesting but don’t have meaning beyond descriptions.

There should be some type of figure where the different categories of degradation are combined with standard errors. Just showing raw data for each site is not particularly useful.

Not sure why rarefaction curves are used to display diversity measures rather than bar graphs, ordinations, etc.

How was the tree in Fig 8 constructed? How are there so few branches on a tree for the soil microbiome?

Figs 8-11 do not seem to really link to the study overall, particularly 11, where the different degradation gradients are not included in the network analysis. There are not hypotheses linked to them, nor are they really discussed in either the results or discussion section.

The raw data is not raw data, just summaries of different measures (diversity, etc.). It seems like raw data here would be sequences (or link to accession number where sequences were deposited). The raw OTU table with taxonomy was perhaps closest to raw data.

Raw sequences provided in supplemental, but should be deposited to a publicly available database where they can be better utilized by a wider audience.

---

## Round 0.2 · Minor Revisions

Please address the reviewers' comments thoroughly. We look forward to receiving your revised manuscript.

Reviewer 1 ·

Basic reporting

Additional language would be great.

Experimental design

1. While the authors claimed that they have added additional references to the method section. Several softwares do not have any reference as claimed.

2. Sequences must be submitted to NCBI SRA or public database.

Validity of the findings

no further comment

Reviewer 3 ·

Basic reporting

The revision work was well carried out following the first evaluation of the manuscript by the reviewers. This improvement of the manuscript highlights some new points that need to be corrected.

English: direct style, easy to understand

References: adapted and relevant

Article structure OK, I have a comment about supplementary material
The data presented in supplemental data are raw data. For this data to effectively enlighten the reader, it would be necessary to present data at a higher level of analysis and only keep the elements which provide additional precision for the discussion in particular. In my opinion, it is not useful to provide all the data, even if this allows the publication to be "redone".

Interesting description of bacterial communities response to environmental constraints

Experimental design

Research in the Aims and scope of PeerJ

Research question very clearly described

Investigation OK

Material and methods
L119-120: The main soil types are brown calcic soil and gray desert soil. Please specify soil classification used.

2.2 Sampling
Close to a drinking point, degradation is mainly due to animal trampling and not grazing. Visually we can interpret the degradation of vegetation as continuous, but the causes are different. Under these conditions, this cannot be interpreted as a gradient.

Where have the sequence been deposited authors have to provide accession number for the sequence dataset.

Validity of the findings

no comment

Additional comments

The revision work was well carried out following the first evaluation of the manuscript by the reviewers. This improvement of the manuscript highlights some new points that need to be corrected.

Degradation gradient is not appropriate and has to be changed. Authors work on sites characterized by different degree of degradation. Different tools and metrics have been used to characterized the degree of degradation of each site. The degree of degradation is rather complex to analyze and a gradient would mean a continuity of the degradation degree. As it cannot be the case, please modify.

Introduction
L62: Zhang Bin et al. (year?) Investigated
L66: Wu Yongsheng et al. (year?)
L89: Hu et al. (year?) found
L95: different degradation gradients degree

The compartment studied must be clearly described: soil or rhizospheric soil and at what depth, fixed or variable following the roots.

---

## Round 0.3 · accepted · Accept

The authors have addressed all the comments and concerns by the reviewers. I have gone through the response letter and the manuscript and assessed the responses and revisions. The manuscript is now ready for publication.